# Silymarin Encapsulated Liposomal Formulation: An Effective Treatment Modality against Copper Toxicity Associated Liver Dysfunction and Neurobehavioral Abnormalities in Wistar Rats

**DOI:** 10.3390/molecules28031514

**Published:** 2023-02-03

**Authors:** Tuba Maryam, Nosheen Fatima Rana, Sultan M. Alshahrani, Farhat Batool, Misha Fatima, Tahreem Tanweer, Salma Saleh Alrdahe, Yasmene F. Alanazi, Ifat Alsharif, Fatima S. Alaryani, Amer Sohail Kashif, Farid Menaa

**Affiliations:** 1School of Mechanical & Manufacturing Engineering (SMME), National University of Sciences and Technology (NUST), Islamabad 44000, Pakistan; 2Clinical Pharmacy Department, College of Pharmacy, King Khalid University, Abha 61441, Saudi Arabia; 3Department of Biology, Faculty of Science, University of Tabuk, Tabuk 71491, Saudi Arabia; 4Department of Biochemistry, Faculty of Science, University of Tabuk, Tabuk 71491, Saudi Arabia; 5Department of Biology, Jamoum University College, Umm Al-Qura University, Makkah 21955, Saudi Arabia; 6Department of Biology, College of Science, University of Jeddah, Jeddah 21589, Saudi Arabia; 7Departments of Internal Medicine and Nanomedicine, California Innovations Corporation, 9, San Diego, CA 92037, USA

**Keywords:** silymarin, liposome nanoparticles, copper toxicity, Wilson’s disease, animal model, histological examination, serological indices, cognitive impairment

## Abstract

Wilson’s disease causes copper accumulation in the liver and extrahepatic organs. The available therapies aim to lower copper levels by various means. However, a potent drug that can repair the damaged liver and brain tissue is needed. Silymarin has hepatoprotective, antioxidant, and cytoprotective properties. However, poor oral bioavailability reduces its efficacy. In this study, a “thin film hydration method” was used for synthesizing silymarin-encapsulated liposome nanoparticles (SLNPs) and evaluated them against copper toxicity, associated liver dysfunction and neurobehavioral abnormalities in Wistar rats. After copper toxicity induction, serological and behavioral assays were conducted to evaluate treatment approaches. Histological examination of the diseased rats revealed severe hepatocyte necrosis and neuronal vacuolation. These cellular degenerations were mild in rats treated with SLNPs and a combination of zinc and SLNPs (ZSLNPs). SLNPs also decreased liver enzymes and enhanced rats’ spatial memory significantly (*p* = 0.006) in the diseased rats. During forced swim tests, SLNPs treated rats exhibited a 60-s reduction in the immobility period, indicating reduced depression. ZSLNPs were significantly more effective than traditional zinc therapy in decreasing the immobility period (*p* = 0.0008) and reducing liver enzymes, but not in improving spatial memory. Overall, SLNPs enhanced oral silymarin administration and managed copper toxicity symptoms.

## 1. Introduction

Copper buildup in the liver and extrahepatic organs is the hallmark of Wilson’s disease (WD), which is characterized as an autosomal recessive copper transport disorder due to decreased copper excretion into the bile [1]. It is a condition caused by a genetic deficiency in the copper-transporting ATPase ATP7B, which is essential for biliary copper outflow and copper loading of ceruloplasmin [2]. The gene (ATP7B), located on chromosome 13, is highly expressed in the liver, kidney, and placenta [3]. With an estimated worldwide allele frequency of 1:90, WD is therefore the most widely recognized form of copper toxicity [4].

The fundamental daily copper need of the body is around 1–2 mg, which is satisfied by dietary copper consumption. Copper is absorbed by intestinal cells and stored safely alongside metallothionein. Copper-transporting ATPase 1 (ATP7A) delivers this copper to the blood [5]. A copper-binding protein known as ceruloplasmin, formed by ATP7B’s transfer of copper to apoceruloplasmin, transports a large portion of the copper in plasma and supplies it to the major organs, including the brain and kidneys [6,7]. Patients with WD often have low ceruloplasmin levels due to the reduced conversion of apoceruloplasmin owing to ATP7B mutations. As a result, copper accumulates in the hepatocytes due to a lack of excretion into the biliary canaliculi [8,9]. Extrahepatic tissues such as the brain and corneas are also affected by excess copper alongside the liver, leading to cirrhosis. Vitamin E levels are lowered in individuals with WD, suggesting that copper has a role in oxidative damage [10]. The liver is the first organ to accumulate copper when it enters the bloodstream; hence, it is primarily affected by chronic copper toxicity. Toxic effects of copper include liver cirrhosis and hemolytic anemia, and injury to the renal tubules, the brain, and other organs [11,12].

Excessive copper may cause membrane lipid peroxidation due to the lipid radicals combining with oxygen to create peroxy radicals [13,14,15]. Hepatotoxicity may occur due to decreased cytochrome c oxidase activity and impaired mitochondrial respiration in the liver due to copper excess [10]. The toxic buildup of copper in the liver, brain, cornea, and kidney causes the symptoms of WD. Bradykinesia, rigidity tremor, dystonia, dysarthria, ataxia, and less frequently, chorea and seizures, are all neurologic manifestations. Hepatic signs include acute and chronic hepatitis, cirrhosis, and hepatic failure, whereas psychological manifestations include depression, personality disorders, cognitive impairment and infrequently, psychosis [3].

In order to prevent or reverse the toxic effects of copper, pharmacological treatment for WD aims to reduce the amount of copper being absorbed, induce the production of endogenous cell proteins, promote the excretion of copper in the urine or bile, or a combination of these [16]. D-penicillamine, trientine, Dimercaprol, and tetrathiomolybdate (chelating agents) are all examples of pharmacological drugs that remove copper. An increase in zinc promotes enterocyte metallothionein, which inhibits copper absorption. For long-term maintenance, zinc treatment is the safest and simplest regimen [17]. Most individuals with abnormal liver function recover after six to twelve months of treatment with medical therapy for this condition. However, therapy must continue throughout the patient’s life [18].

Liver disorders have been treated for years using milk thistle (*Silybum marianum* L.), an old medicinal plant from the Carduus marianum family [19]. Milk thistle extract, also known as silymarin, has a remarkable biological impact due to the active component called silybin [20]. Silymarin has been discovered to be cytoprotective, antioxidant, and hepatoprotective in the medical community [21]. Because it scavenges free radicals and boosts glutathione levels, silymarin may be used to treat hepatitis, hepatic cirrhosis, and mushroom poisoning, and it has antifibrotic, immunomodulating, and anti-inflammatory benefits. Lipid peroxidation and hepatocyte external cell membrane alteration are the two mechanisms of action of silymarin that reduce the oxidative stress generated by toxins that disrupt cell membranes [22]. Silymarin works as an antioxidant not only because it functions as a free radical scavenger, but also because it has an effect on enzyme systems related to glutathione and superoxide dismutase [23]. Because of its antioxidant, anti-inflammatory, and antifibrotic properties, silymarin is one of the most often utilized natural compounds to treat liver diseases [24].

It has been discovered that all components of silymarin can inhibit lipid peroxidation. It protects the mitochondria and microsomes in rat liver cells against chemicals responsible for lipid peroxide generation in vitro [25,26,27,28]. Silymarin is reported to increase the production of ribosomes, DNA synthesis, and protein synthesis; it stimulates RNA polymerase I and the transcription of rRNA in the nucleus, increasing ribosomal protein production in the cell membrane and the nucleus, respectively [29,30]. Restoration of hepatotoxin-damaged structural proteins and enzymes happens, resulting in stimulated protein synthesis and healing [31].

Apart from all aforementioned benefits, the low oral bioavailability of silymarin limits its potential as a therapy for inflammation. The quick metabolism and elimination, together with its poor water solubility and limited permeability through intestinal epithelial cells, are the primary reasons that restrict its bioavailability [32]. Both factors contribute to poor absorption. Furthermore, high molecular weight contributes to its low bioavailability [33]

This study aims to evaluate the beneficial properties of silymarin against copper toxicity, the phenotypic pathological condition of WD, and whether it can be used as a supportive therapy for such patients. To enhance the bioavailability, liposomal nanocarriers were synthesized in which silymarin was incorporated for better oral administration. Liposomal nanoparticles (LNPs) are a promising delivery system for drugs and other substances that are difficult to deliver to the liver and can be quickly taken up and delivered to specific areas within the liver. The liver has a high concentration of scavenger receptors, which are proteins that bind to and absorb LNPs [34]. Furthermore, PEGylation of SLNPs was carried out to extend the blood circulation time and penetrate the blood–brain barrier (BBB) [35]. To check the efficacy of these SLNPs, a copper toxicity model was established in Wistar rats.

## 2. Results

### 2.1. Characterization of SLNPs and BLNPs

Both SLNPs and Blank liposome nanoparticles (BLNPs) were successfully synthesized, as shown by physical characterization.

#### 2.1.1. Ultraviolet–Visible (UV-VIS) Absorption Spectroscopy

UV–VIS absorption spectroscopy of silymarin medication exhibited the surface plasmon resonance (SPR) peak primarily at 230 nm, BLNPs at 207 nm, and SLNPs at 206 nm. The peaks of cholesterol, dipalmitoylphosphatidylcholine (DPPC), and polyethylene glycol (PEG) 2000, were found at 204 nm, 202 nm, at 205 nm, respectively (Figure 1). Characteristic peaks of SLNPs corresponds to their components which confirmed the synthesis of SLNPs.

#### 2.1.2. Fourier Transform Infrared Spectroscopy (FTIR) Analysis

The FTIR spectrum of cholesterol indicated peaks or bands at 2850 cm^−1^ (CH stretch, alkanes), 873 cm^−1^ (tri-substituted aromatics). DPPC spectrum indicated peaks at 2919/cm (CH stretch, alkanes), 1632 cm^−1^ (R-NH2, amines), 1115 cm^−1^ (C-O stretch, ether), 720 cm^−1^ (RCH2CH3, bending mode). PEG-2000 spectrum delineated peaks at 3429 cm^−1^ (O-H stretch, alcohol), 2923 cm^−1^ (CH stretch, alkanes) and 1638 cm^−1^ (C=C stretch, alkenes). The observed changes in infrared bands proved the conformational changes in lipid biomolecules by incorporating with the silymarin drug and PEG 2000, which masked the peaks due to coating, as seen in Figure 2.

#### 2.1.3. Particle Size

The scanning electron microscopy (SEM) was used to determine the size of the SLNPs and BLNPs. It was revealed that the SLNPs exhibited spherical morphology and had an average size of 67 nm, as seen in Figure 3, whereas the BLNPs had an average size of 24 nm. This demonstrated successful loading of silymarin in the liposome [36].

#### 2.1.4. Zeta Potential

The average zeta potential of SLNPs and BLNPs were −14.8 mV and −17.8 mV, respectively, as seen in Figure 4a,b.

#### 2.1.5. Drug Encapsulation Efficiency and Drug Loading Capacity

The encapsulation efficiency of SLNPs was 74%. The drug loading capacity was found to be 47.6%.

#### 2.1.6. Drug Release Kinetics

The software ‘KinetDS’ was used to determine the release kinetics (Figure 5). Zero order kinetics, as seen in Figure 5b, were observed which depicted drug released from the SLNPs maintained a steady release throughout the time.

### 2.2. Treatment of Copper Toxicity

#### 2.2.1. Forced Swim Test

There was a significant increase in immobility time of diseased rats as compared to the normal group from 130 to 215 s (*p* = 0.0002), as seen in Figure 6. Treatment with SLNPs significantly decreased this time to 155 s as compared to no treatment (*p* = 0.0004). SLNPs also performed significantly better than silymarin (*p* = 0.002). The combined treatment of ZSLNPs decreased immobility time to 167 s which is significantly less than the Zn group which had an immobility period of 210 s (*p* = 0.0008). Hence, treatment of the diseased rats with SLNPs significantly mitigated depression symptoms.

#### 2.2.2. Y Maze Test

There was a significant decrease in % spatial memory of diseased rats as compared to the normal group from 82% to 49% (*p* = 0.0009). Treatment with SLNPs significantly increased spatial memory to 70% as compared to no treatment (*p* = 0.006). SLNPs also performed significantly better than silymarin which only increased the spatial memory to 55% (*p* = 0.023), as seen in Figure 7. Hence, treatment of the diseased rats with SLNPs significantly improved spatial memory, which is a marker of cognitive functioning.

#### 2.2.3. Body and Liver Weights

At the beginning of the experiment, during the acclimatization period, the weight of the rats normally increased due to growth, as seen in Figure 8a. During the period of disease induction, the rate of weight increase was slower, and the weight decreased in some rats. There was a significant decrease in the body weight of diseased rats compared to the normal group at the end of the experimental period (*p* = 0.017). The average weight for this group was 201 gm before disease induction, and after the copper exposure had been stopped, the rats had an average weight of 182 gm. With the treatment of SLNPs, the weight increased to 202 gm. There was a significant effect of SLNPs on decreasing the liver weight from 11.4 gm to 10.4 gm (*p* = 0.021), compared to the diseased group. The rest of the treatments did not exhibit any significant changes in liver weights, as seen in Figure 8b.

#### 2.2.4. Serological Analysis (Liver Function Tests)

The total bilirubin (T.B) levels were significantly increased in the diseased group (*p* = 0.022) and BLNPs group (*p* = 0.024) as compared to the normal rats from 0.1 to 0.14 mg/dL, as seen in Figure 9a. With treatment with SLNPs, the T.B levels decreased significantly from 0.14 to 0.09 mg/dL as compared to the diseased group (*p* = 0.002). SLNPs treatment also showed better results of bilirubin than silymarin treatment group in which the rats had a 0.12 mg/dL bilirubin (*p* = 0.011), and the combination therapy of ZSLNPs exhibited lower T.B than Zn therapy (*p* = 0.072). There was a significant increase in aspartate transaminase (AST) levels of diseased rats as compared to the normal group from 188 to 518 U/L (*p* = 0.00006), as seen in Figure 9b. Treatment with SLNPs significantly decreased AST to 281 U/L as compared to no treatment (*p* = 0.0003). SLNPs also decreased AST levels significantly better than silymarin in decreasing AST (*p* = 0.0014) and the combined treatment of ZSLNPs decreased AST levels significantly to 232 U/L as compared to zinc therapy which had 473 U/L AST (*p* = 0.00007).

There was a significant increase in alanine transaminase (ALT) and alkaline phosphatase (ALP) levels of diseased rats as compared to the normal group (*p* = 0.0002, *p* = 0.0002), as seen in Figure 10a,b, respectively. Treatment with SLNPs significantly decreased ALT and ALP as compared to no treatment (*p* = 0.0007, *p* = 0.0083). SLNPs also performed significantly better than silymarin in decreasing the two enzymes (*p* = 0.0011, *p* = 0.0456) and the combined treatment of ZSLNPs decreased ALT and ALP levels significantly as compared to Zn therapy (*p* = 0.0085, *p* = 0.0003).

#### 2.2.5. Hepatic and Brain Histopathology

The histological staining of the different liver cells is depicted in Figure 11. The histological examination of the livers of the control rats showed that the hepatocytes, portal triads, and vasculature were all normal. Massive fatty change and centrilobular necrosis were seen in the livers of diseased rats as well as those who had been treated with BLNPs. In addition to this, it was found that several of the rats had instances of necrosis in their hepatic cells. As can be observed in Figure 11b,c, both of these groups had signs of hepatitis, which are defined by the infiltration of mononuclear cells—mostly macrophages and lymphocytes—around the major veins and in the portal sections of the liver. In addition, the diseased rats’ hepatocytes displayed diffuse cytoplasmic vacuolation of the hydropic type. In this condition, the cells were swollen as clear fluids, and the nucleus replaced the cytoplasm. The diseased rats’ hepatocytes also displayed moderate sharp outline vacuoles of the fatty change type alongside hepatocellular necrosis with extravasated red blood cell and karyolysis, which is the complete breakdown of the chromatin of a dying cell due to the enzymatic destruction by endonucleases, was seen in the diseased and BLNPs group. Karyolysis occurs when a cell dies, and its chromatin is destroyed by endonucleases. The cells in these specimens are shown to have swollen, which is accompanied by the breakdown of the cell membrane and the spilling of cytoplasm into the surrounding tissue. In these necrotic cells, the nuclei are either hardly visible or totally missing.

The normal group was given a histopathological score of 1 as the cellular degeneration alterations were nonexistent. The number of hepatic cells with granular and vacuolar degeneration was increased in diseased and BLNPs treated group. Hence, both these groups were scored as 5, with the cellular degeneration alterations being very severe and extensive. The number of such abnormal cells was slightly decreased in the silymarin, and zinc treated groups (Figure 11d,f). These two groups were scored as 3, with the cellular degeneration alterations being moderate. This degeneration decreased significantly in SLNPs group and ZSLNPs group (Figure 11e,g). Both these groups had a high number of cells with normal morphology, and they displayed nearly normal histoarchitecture. These two groups were thus scored 2, with the cellular degeneration alterations being mild.

The histological staining of the different brain cells is depicted in Figure 12. Histopathological alterations in control rats did not deviate from the norm. In the diseased and BLNPs-treated rat brain areas, abnormalities including cell swelling, neuronal vacuolation, demyelination, neuronophagia, and settelitosis were found. The appearance of the neural cells in these groups was either globular or spindle shaped. In the diseased and BLNPs groups, it was frequent to see degraded neurons, dilated blood vessels, vacuolated foci with cellular loss, high eosinophilic staining of neurons, inflammatory cells infiltration, and occasional vacuolation of neurons (Figure 12b,c). Apoptosis was suggested by the presence of shrunken neurons with pyknotic nuclei and limited amounts of eosinophilic cytoplasm in the cells. In addition, a significant rise in the total number of glial cells was seen in these groups. It was also noted that nodule development, proliferation, and gliosis took place. It was also discovered that the myelin sheath had degenerative alterations. In addition, fibrosis, proliferation of connective tissue, and infiltration of inflammatory cells into the periventricular system were all seen in the brain tissue of these groups. In the ventricles, there was ependymal cell hyperplasia, and there was also periventricular inflammation, which was characterized by the growth of fibrous tissue and the infiltration of inflammatory cells.

These pathologies improved in the treatment groups of SLNPs and ZSLNPs. The normal group was given a histopathological score of 1 as the cellular degenerative alterations were nonexistent. The number of neural cells with granular and vacuolar degeneration was increased in diseased BLNPs treated group. Hence, both these groups were scored as 5, with the cellular degeneration and inflammatory alterations being very severe and extensive. The number of such abnormal cells was slightly decreased in the silymarin, and zinc treated groups (Figure 12d,f). These two groups were scored as 3, with the cellular degeneration and inflammatory changes being moderate. This degeneration and inflammation decreased significantly in the SLNPs group and ZSLNPs group, as seen in Figure 12e,g, respectively. Both these groups had a high number of neural cells with normal morphology, and they displayed nearly normal histoarchitecture. These two groups were thus scored 2, with the degeneration and inflammation being mild.

## 3. Discussion

In this study, silymarin and its liposomal formulation were investigated for their role in the treatment of liver dysfunction and cognitive impairment caused by copper toxicity. Silymarin was selected owing to its antioxidant, anti-inflammatory, and antifibrotic properties, as well as its role in the treatment of a variety of liver ailments, including cirrhosis, hepatocellular carcinoma, and chronic liver disease [37]. By counteracting oxidative stress, insulin resistance, liver fat buildup, and mitochondrial dysfunction, silymarin can be a useful medication for the treatment of liver diseases [22].

The aim of this study was to determine whether a liposomal formulation of silymarin is effective as a supportive medication to ameliorate the symptoms of copper toxicity in WD. The study demonstrated SLNPs to be an effective treatment strategy to improve cognitive functioning, reduce depression symptoms, and reverse liver dysfunction. Silymarin, due to its hydrophobic nature, was successfully encased in the bilayer of a liposomal vesicle. Whereas, PEG coating provided a stealth effect to enhance circulation time and bioavailability [38]. Nanoparticle size was measured to be 24 nm for unloaded liposomes and 67 nm for silymarin-loaded liposomes. The size increase confirmed the drug loading [36,39]. However, zeta potential measurements showed values of −14.8 mV and −17.8 mV. Alterations in the zeta potential are often used as an additional measure of the extent of medication that has been loaded into nanoparticles. In order to maintain stability and prevent the aggregation of particles, it is required to have higher zeta potential values, whether positive or negative [40]. Therefore, a rise in the charge of the SLNPs from −14.8 to −17.8 indicates an improvement in the liposome’s stability. The small size of SLNPs and structure based on lipids was comparable to the phospholipid bilayer that is found in the blood–brain barrier [35].

In the current study, copper sulphate was used to induce a copper toxicity model, as it was proved to be very effective [41] The reason behind this is its poisonous nature at high concentrations that may lead to cellular damage [42]. Increased levels of copper in the blood have been linked to poor memory function in a number of studies on both humans and animals, suggesting that the metal may play a role in these pathologies through its direct or indirect effects on the neurotransmission responsible for cognitive abilities, including the cholinergic, glutamatergic, and GABAergic systems [43]. Furthermore, prolonged exposure to Cu causes oxidative stress in the CNS, which is among the most impacted systems, particularly in the hippocampus, a part of the brain that is crucial for memory function [44].

The present study reported increased depression-like symptoms in rats due to copper toxicity as observed by the increase in immobility periods of the diseased rats. Similar effects were reported in a previous study where the immobility period of rats that were administered with copper sulfate for 6 weeks was significantly increased from 50 s to almost 80 s as compared to naïve rats [45]. The effects of copper poisoning on cognitive performance were also investigated. It was observed that the spatial memory of the rats decreased as well, suggesting an impairment in their cognitive ability. Similar effects were observed in previous studies where the Y maze test results revealed lower percentage spatial memory in rats exposed to copper [41,43,46].

It is believed that hepatocytes are one of the primary cell types affected by Cu poisoning [47]. Several investigations have shown that Cu causes hepatocytes to undergo apoptosis, mitochondrial abnormalities, and oxidative stress [48]. One common mechanism to explain the cellular toxicity of Cu is that it may cause oxidative stress by increasing reactive oxygen species (ROS) production [16,49]. Keeping this in view, the effects of copper were examined on liver and brain by histological examination combined with liver function testing. There was an increase in liver enzymes which corresponds to liver disease. In the present study, the diseased group had higher T.B, AST, ALP, and ALT levels as compared to the normal group. The hepatic toxicity was evident by these results and upon treatment through SNLPs and ZSLNPs, these T.B and transaminases levels decreased significantly. Kumar et al. (2014) also reported increased liver enzyme levels in rats that were administered copper sulfate in the doses 100 and 200 mg/kg BW. The latter group had higher liver enzyme levels than the group given the lower dose [41]. In another study, the role of components of silymarin decreased plasma levels of AST, ALT, and gamma-glutamyl transpeptidase (GGT) following paracetamol intoxication [26].

Histological examination proved these detrimental effects of copper on the liver and brain cells of the rats. There was significant degeneration in both the hepatocytes and neurons that correspond to the phenotypic manifestations of copper toxicity [12,50]. The toxicity observed in histological examination was attributed to the ROS and oxidative stress that may have triggered apoptosis [42]. Overproduction of ROS could have been induced by high Cu concentrations, further affecting mitochondrial electron transport, which may have eventually led to mitochondrial impairment and apoptosis [51,52].

## 4. Materials and Methods

### 4.1. Materials

The materials were purchased from Sigma-Aldrich unless indicated otherwise. Female Wistar rats were purchased from Atta-ur-Rahman School of Applied Biosciences (ASAB), National University of science and technology (NUST).

### 4.2. Synthesis of PEGylated Liposome Nanoparticles

A total of 1.6 mL of 0.5 mM DPPC and 0.4 mL of 0.5 mM cholesterol were mixed with 8 mL ethanol to make a final solution of 10 mL. An amount of 0.1 mL of a 1 mM silymarin solution was mixed in the lipid solution to make SLNPs. BLNPs were also synthesized in which no drug was loaded, but the same procedure was followed. The solution was bath sonicated for 40 min. A total of 10 mL of the lipid and water phases were then heated up to 60 °C in a water bath. The lipid solution was combined with deionized water (DI) and probe sonicated for 40 min. Rotary evaporation above the phase transition temperature (50 °C) was carried out to remove ethanol from this solution. DI water was added to dilute the nanoparticle mixture to 50 mL. A total of 20 mL of 0.25% PEG 2000 solution was dropped into the nanoparticle mixture while stirring. Rotary evaporation was then repeated until 10 mL DI water was left [39].

### 4.3. Characterization of Nanoparticles

Multiple characterization techniques were conducted to assess particle size, shape, surface charge, drug encapsulation, and release efficiency of PEGylated SLNPs [53].

#### 4.3.1. U.V-Vis Absorption Spectroscopy

A UV-Vis 2800 (BMS Biotechnology Medical Services, Madrid, Spain) was used to analyze the U.V.-Vis spectra of SLNPs and BLNPs from 200–600 nm.

#### 4.3.2. Fourier Transform Infrared Spectroscopy (FTIR) Analysis

Liquid samples for the nanoparticles were used for FTIR analysis. Using a Bruker FTIR Spectrophotometer ALPHA II (Westborough, MA, USA), FTIR spectra were obtained between 4000–650 cm^−1^.

#### 4.3.3. Particle Size

Using the VEGA3 LMU Scanning Electron Microscope (Tescan, Brno, Czech Republic), particle size and morphology were evaluated.

#### 4.3.4. Zeta Potential

Using Malvern Zeta Sizer Version 7.12 (Malvern, Worcestershire, UK), the zeta potential (surface charge) of both types of LNPs was determined.

#### 4.3.5. Drug Encapsulation Efficiency

Efficacy defines the quantity of medication encapsulated in the liposome vesicles. A UV spectrophotometer at 330 nm absorbance was used to test several dilutions of the medication to generate a feasible linear standard curve for determining drug encapsulation efficiency. It was possible to arrive at the equation Y = mx + c. The *unentrapped* silymarin was then calculated using this standard curve value. This was followed by plugging the results into the following formula [54,55].
(1)$$\mathrm{Encapsulation}\;\mathrm{Efficiency}\;\left(\%\right)=\frac{Total\;Drug-Unentrapped\;Drug\;}{Total\;Drug}\times 100$$

#### 4.3.6. Drug Release Efficiency

With the addition of a certain amount of phosphate-buffered saline (PBS), the drug release of SLNPs was evaluated for up to 15 h. Centrifugation was carried out, and the supernatant was analyzed by UV the spectrophotometer. At a wavelength of 230 nm, absorbance values were measured and utilized as a measure of cumulative drug release. A BLNPs solution was used as a control for the whole investigation [56].

### 4.4. Development of Copper Toxicity Model

#### 4.4.1. Animals

For this experiment, 5–7 week old female Wistar rats (*n* = 35) were employed, each weighing 80–120 g. Separate cages for each treatment group (*n* = 5) with water and food were used to house the rats, maintained on a 12-h light/dark cycle, 27 °C temperature and 60–70% humidity. The rats were acclimatized for 7 days before the induction was started. Good laboratory practices given by the US Food and Drug Administration (FDA) in 2010 dictated how rats were cared for and handled.

#### 4.4.2. Copper Toxicity Induction

The rats were randomly allocated into 7 groups of 5 rats assigned per group. One group was the positive control (PC) in which the rats were not given any chemicals throughout the study. The rest of the 6 groups were given copper sulfate at a dose of 200 mg/kg body weight daily for up to 90 days through oral gavage [41].

### 4.5. Treatment Design

The antioxidant properties of SLNPs were examined in experiments with diseased rats. The 30 rats were divided in 6 groups of 5 rats each, based on the treatment plans. The treatment groups were as follows: diseased (untreated), silymarin treated group with a dose of 10 mg/kg body weight (BW), SLNPs treated group with a dose of 500 µg/kg BW, BLNPs treated group with a dose of 500 µg/kg BW, zinc sulfate treated group (Zn) with a dose of 10 mg/kg BW, and ZSLNPs with doses of 10 mg/kg BW zinc sulfate and 500 µg/kg BW SLNPs. All treatments were administered for a period of 21 days, biweekly, through oral gavage. After treatment, tests for anxiety and spatial memory were conducted (Figure 13).

#### 4.5.1. Forced Swim Test

A cylindrical tank filled with 24 ± 1 °C water was utilized for the forced swim test (FST). Each subject was put in the tank alone, and immobility time was measured for six minutes. The subject was immobile if it ceased struggling and remained still while afloat or only made the minimal movements required to maintain its head above water. The subjects were then taken out of the water, dried with a towel, and returned to their cages [45].

#### 4.5.2. Y Maze Test

The Y maze test was used to examine subjects’ working memory by observing their spontaneous alternation behavior. A Y-shaped apparatus with three arms, (45 cm × 35 cm × 12 cm) was utilized. Rats were placed at the intersection of three arms and were allowed to make arm choices for six minutes. The *percentage of alterations* was calculated using the following equation [43]:(2)$$Percent\;Alteration=\frac{Spontaneous\;Alteration}{Total\;number\;of\;arm\;entries-2}\times 100$$

#### 4.5.3. Body and Liver Weights

Throughout the study, each rat was weighed using a tared weighing balance at the beginning of each week. The final body weights were recorded after the treatment period had ended.

After performing the tests, rats were anesthetized with chloroform before being sacrificed. Rat livers and brains were collected, and weights were recorded. The organs were preserved in a 10% neutral-balanced formalin solution. They were then embedded in the paraffin.

#### 4.5.4. Serological Indices

AST, ALP, ALT and T.B tests were performed on blood collected from the heart.

#### 4.5.5. Histological Examination

A measure of 5 µm serial slices of the paraffin embedded liver and brain were obtained, and structural alterations in organs were observed using hematoxylin and eosin (HE) staining on histological slides. Utilizing a microscope, an assessment of the breadth of the pathological alterations was carried out. Every deviation from the typical structure was meticulously documented. The grading standards were used for rating the histology of the liver and the brain. Vascular, inflammatory, and cellular degenerations were each given a score between 1 and 5, with 1 being nonexistent, 2 being mild, 3 being moderate, 4 being severe, and 5 being very severe (extensive) [57].

#### 4.5.6. Statistical Analysis

Statistical analyses were performed using GraphPad Prism (Version 8.0, San Diego, CA, USA). Mean and standard deviations were reported. Furthermore, group comparisons were carried out using independent *t*-test. 5 samples from each group were taken and compared with another group using independent *t*-test to find out the statistical significance between the two groups being compared. These comparisons were carried out for pairs of different groups such as Normal: Diseased, Diseased: SLNPs, SLNPs: Silymarin, and Zn: ZSLNPs. The *p* values were recorded as such and values lower than the alpha value of 0.05 were chosen as statistically significant.

## 5. Limitations

Additional experiments concerning neurological functions, as well as liver and brain redox states, should be performed to confirm the obtained results. Measurement of copper levels in the brains and livers can also be performed to check accumulation and deposition levels.

## 6. Conclusions

SLNPs proved to be an effective treatment for managing the phenotypic effects of copper toxicity. SLNPs significantly reduced liver enzymes AST, ALP, ALT, and T.B. SLNPs also significantly improved the cognitive behavior of rats along with treating depression symptoms. Hence, SLNPs exhibited good oral delivery of silymarin. As compared to the traditional therapy of zinc, a combined therapy of zinc and supplementary silymarin LNPs proved to be a more effective treatment plan. This in vivo study showed efficiency of the SLNPs against liver dysfunction and neurobehavioral changes caused by copper toxicity. Further investigations to analyze the effects of SLNPs on cytokines and oxidative stress are necessary and will be vital for the further assessment of these novel nanoparticles.

## Figures and Tables

**Figure 1 molecules-28-01514-f001:**
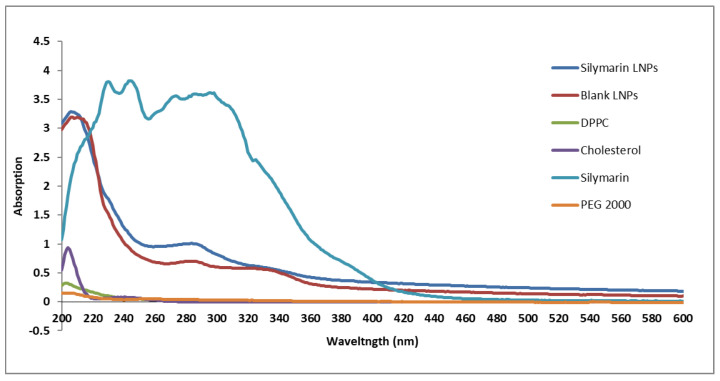
Comparative UV–VIS spectra of blank and silymarin LNPs and their components DPPC, cholesterol, silymarin, and PEG 2000, using a UV–Vis 2800 (BMS Biotechnology Medical Services, Madrid, Spain).

**Figure 2 molecules-28-01514-f002:**
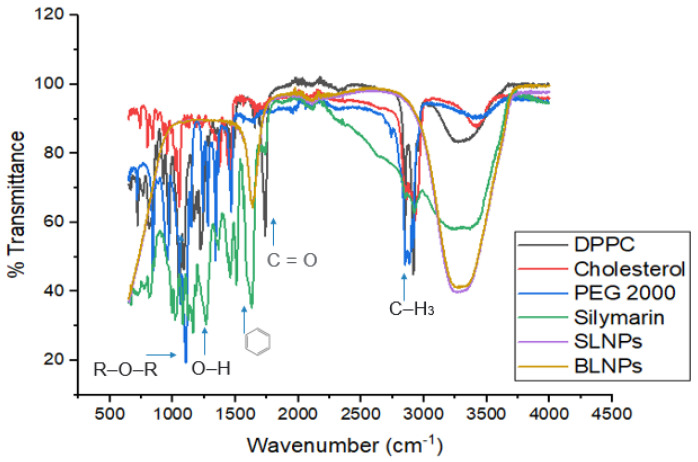
Comparative FTIR spectra of DPPC, cholesterol, PEG−2000, silymarin, silymarin LNPs, and blank LNPs, using a Bruker FTIR spectrophotometer ALPHA II (Westborough, MA, USA).

**Figure 3 molecules-28-01514-f003:**
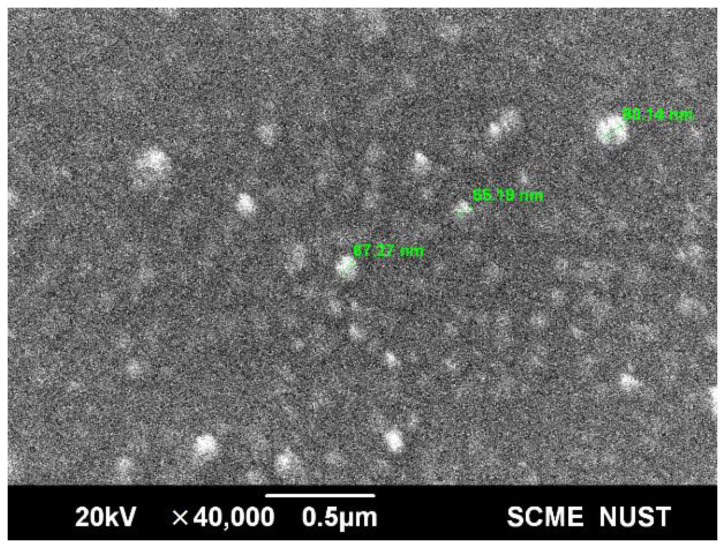
Scanning electron microscopy image of silymarin LNPs, depicting their size and morphology, using the VEGA3 LMU scanning electron microscope (Tescan, Brno, Czech Republic).

**Figure 4 molecules-28-01514-f004:**
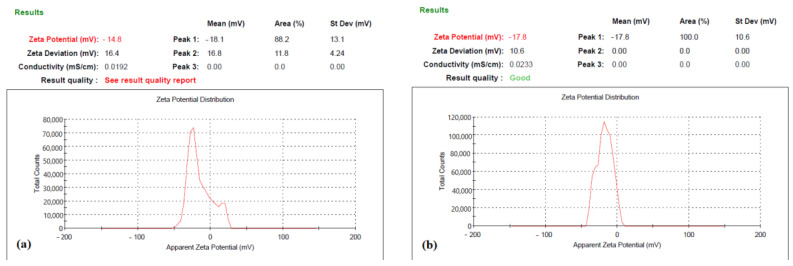
Zeta Potential of (**a**) blank LNPs and (**b**) silymarin LNPs, using Malvern Zetasizer, version 7.12 (Malvern, Worcestershire, UK).

**Figure 5 molecules-28-01514-f005:**
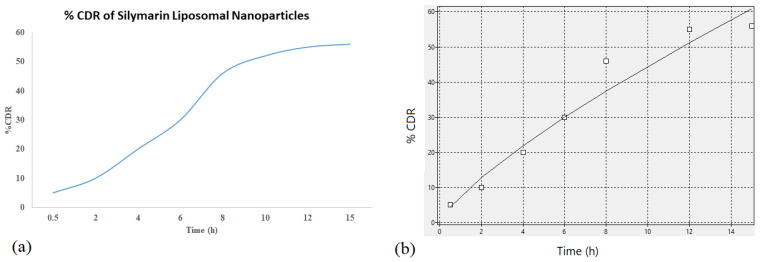
(**a**) Cumulative drug release of silymarin LNPs over 15 h and (**b**) curve fitting using KinetDS software to determine release model using the subplots.

**Figure 6 molecules-28-01514-f006:**
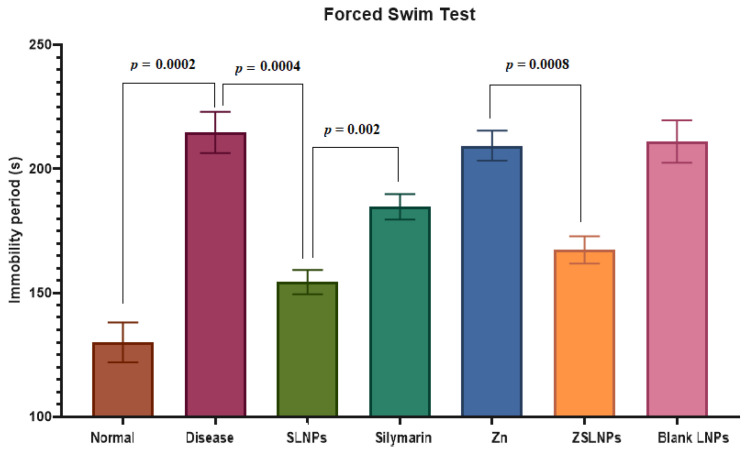
Mean immobility time of rats (*n* = 5), and statistical significance between groups using independent *t*-test, as observed in forced swim test, carried out after the following treatments for a period of 21 days through gavage: diseased, SLNPs (500 µg/kg BW), silymarin (10 mg/kg BW), Zn (10 mg/kg BW), ZSLNPs (10 mg/kg BW + 500 µg/kg BW), and BLNPs (500 µg/kg BW).

**Figure 7 molecules-28-01514-f007:**
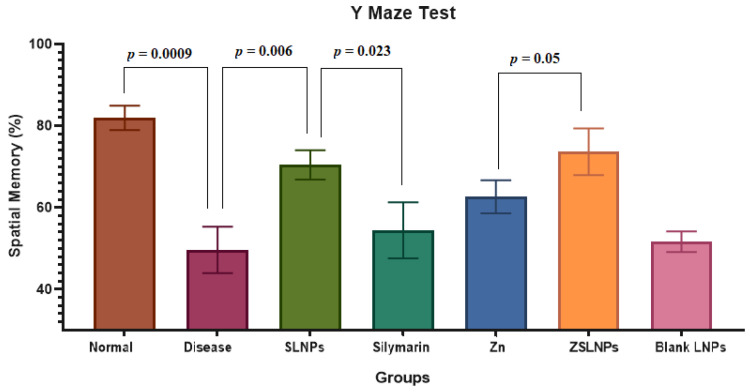
Mean Percentage Spatial Memory of Rats (*n* = 5), as observed in Y maze test and calculated using equation ii, and statistical significance between groups independent *t*-test, carried out after the following treatments for a period of 21 days: diseased, SLNPs (500 µg/kg BW), silymarin (10 mg/kg BW), Zn (10 mg/kg BW), ZSLNPs (10 mg/kg BW + 500 µg/kg BW), and BLNPs (500 µg/kg BW).

**Figure 8 molecules-28-01514-f008:**
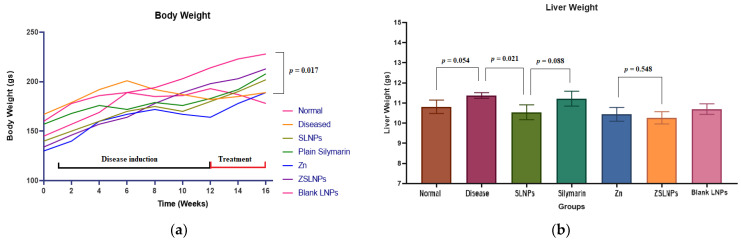
(**a**) Mean body and (**b**) liver weights of rats (*n* = 5), with the body weight recorded weekly and liver weights measured after sacrifice using a weight scale, and statistical significance between groups using independent *t*-test, following the different treatments for a period of 21 days through gavage: diseased, SLNPs (500 µg/kg BW), silymarin (10 mg/kg BW), Zn (10 mg/kg BW), ZSLNPs (10 mg/kg BW + 500 µg/kg BW), and BLNPs (500 µg/kg BW).

**Figure 9 molecules-28-01514-f009:**
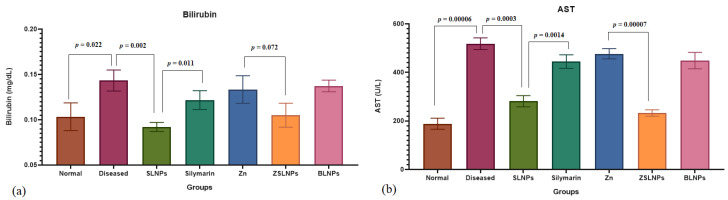
Mean liver function test results of rats (*n* = 5) (**a**) bilirubin and (**b**) AST, and statistical significance between groups using independent *t*-test, after sacrifice following the different treatments for a period of 21 days through gavage: diseased, SLNPs (500 µg/kg BW), silymarin (10 mg/kg BW), Zn (10 mg/kg BW), ZSLNPs (10 mg/kg BW + 500 µg/kg BW), and BLNPs (500 µg/kg BW).

**Figure 10 molecules-28-01514-f010:**
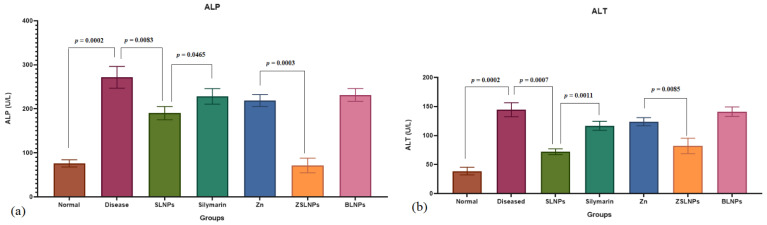
Mean liver function test results of rats (*n* = 5) (**a**) ALP and (**b**) ALT, and statistical significance between groups using independent *t*-test, after sacrifice following the different treatments for a period of 21 days through gavage: diseased, SLNPs (500 µg/kg BW), silymarin (10 mg/kg BW), Zn (10 mg/kg BW), ZSLNPs (10 mg/kg BW + 500 µg/kg BW), and BLNPs (500 µg/kg BW).

**Figure 11 molecules-28-01514-f011:**
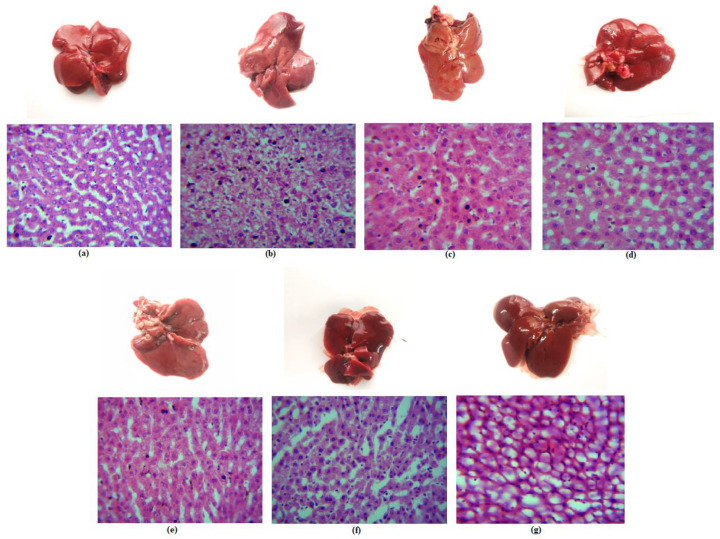
Liver histopathology of rats after sacrifice following the treatments for a period of 21 days (**a**) normal (**b**) diseased (**c**) BLNPs Treated (500 µg/kg BW) (**d**) silymarin treated (10 mg/kg BW) (**e**) SLNPs Treated (500 µg/kg BW) (**f**) Zn Treated (10 mg/kg BW) (**g**) ZSLNPs Treated (10 mg/kg BW + 500 µg/kg BW).

**Figure 12 molecules-28-01514-f012:**
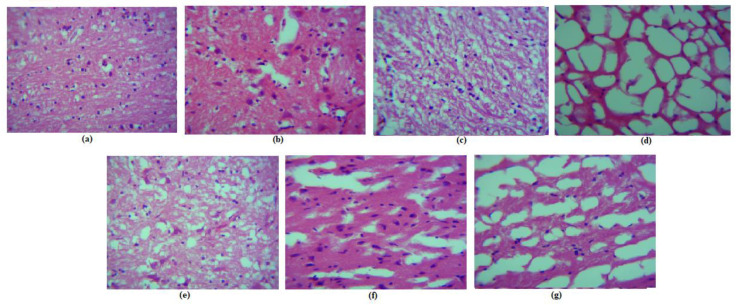
Brain histopathology of rats after sacrifice following the treatments for a period of 21 days (**a**) normal (**b**) diseased (**c**) BLNPs Treated (500 µg/kg BW) (**d**) silymarin Treated (10 mg/kg BW) (**e**) SLNPs Treated (500 µg/kg BW) (**f**) Zn Treated (10 mg/kg BW) (**g**) ZSLNPs Treated (10 mg/kg BW + 500 µg/kg BW).

**Figure 13 molecules-28-01514-f013:**
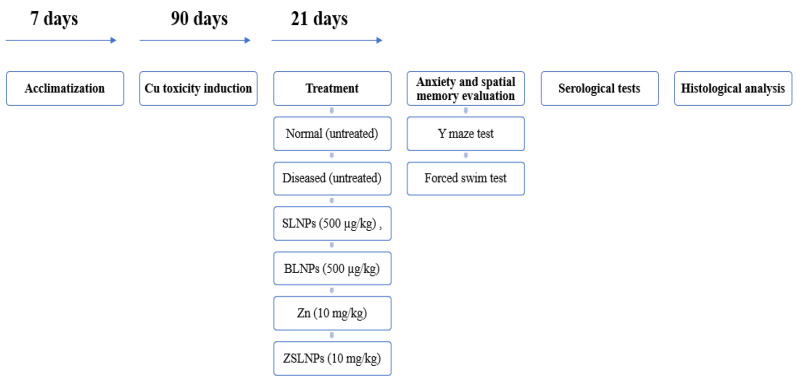
Flow chart showing study design from acclimatization, induction, treatment till analysis.

## Data Availability

Not applicable.

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
