# Peer review of "Silymarin Encapsulated Liposomal Formulation: An Effective Treatment Modality against Copper Toxicity Associated Liver Dysfunction and Neurobehavioral Abnormalities in Wistar Rats"

_molecules, 2023, doi:10.3390/molecules28031514_

Round 1

Reviewer 1 Report

The authors addressed the issue of protective effects of silymarin liposomal nanoparticles (SLNPs) in rat model of Wilson’s disease by assessing the behavior using forced swim test and Y maze test; liver function by testing the levels of AST (Aspartate transaminase or aspartate aminotransferase test), ALP (Alkaline 217 Phosphatase), ALT (Alanine transaminase) and T.B (Total Bilirubin)); and the structural alterations in organs (liver and brain) using Hematoxylin and Eosin (HE) staining. The authors, also, estimated the body and liver weights. All of this was preceded by thorough characterization of the newly synthesized SLNPs. The results indicate that SLNPs are able to ameliorate copper toxicity symptoms by decreasing the liver enzymes and enhancing the rats' spatial memory, reducing the immobility period as an indicator of depression which are valuable observations. However, there are some issues that need to be addressed:

1.  Section Abstract should be more concisely written. For instance, it is stated “Medical researchers have shown silymarin and silibinin to have hepatoprotective, antioxidant, and cytoprotective properties”. What is the importance of silibinin for this section and even the whole manuscript when the authors tested only the effects of silymarin and silymarin liposomal nanoparticles, not silibinin? Since the authors tested the effects of several nanoparticles It is confusing exactly which nanoparticles decreased the liver enzymes and enhanced rats' spatial memory. These are just some examples of the shortcomings of this section. It is necessary to read it thoroughly and rewrite it to be much more fluent and informative.

2.  In section Introduction what is the importance of describing the silibinin effects when the authors tested only the effects of silymarin and silymarin liposomal nanoparticles, not silibinin?

3.     In section Material and methods it is necessary to list the producers/manufacturers of the used chemicals. The experimental groups need to be better described. In subsection 2.4.2. Copper Toxicity Induction, the sentence “The rats were acclimatized for 7 days before the induction was started.” needs to be moved to the subsection 2.4.1. Animals, since it is general statement and applies to all animals. How were the animals kept, 2-4 per cage or individually? If they were individually caged - why the authors choose to do that when it is known that social isolation leads to the depression and/or depression-liked conditions? When were the animals sacrificed? It is not defined when and how the bodies and liver weights were determined. Also, the statistical analysis of the results is not defined. Were the weights of brains different between experimental groups? Subsections need to be logically rearranged since some of the methods were performed before while some methods after the sacrifice. Additional experiments concerning neurological functions, as well as the livers’ and brains’ redox states need to be performed to confirm the obtained results. These are just some examples of the shortcomings of this section and the study in general. It is necessary to read it thoroughly and rewrite it to be much more fluent and informative, and complement the study with the required methods and results.

4.     Section Results is not well written, the presentation of results needs to be uniformed and to be more focused on the biological effects of investigated nanoparticles. It is necessary to improve this section with the results of the requested methods. Each subsection is missing a reference of the corresponding Figure. Moreover, there are two Figures 1, 2, 3 and 4, so the numbers of the Figures must be arranged in appropriate order.

5.   Section Discussion is not thorough enough and the key points are not well discussed. Overall, the paragraphs are too general and are not connected to the investigated nanoparticles’ effects. Thus, it should be improved and more concisely written so the findings and interpretations are better linked to the results of prior published studies, while the obtained/presented results should be discussed in better manner.

6.   Some abbreviations are used before they are introduced and once introduced the abbreviations should be used through the manuscript in the same manner. The authors should read the manuscript thoroughly and uniform this item.

Reviewer 2 Report

Maryam et al. formulated liposomal nanoparticle to deliver silymarin into mice with Wilson’s disease. The encapsulated silymarin reversed liver toxicity associated with copper and neurobehavioral abnormalities. This study is novel as silymarin by itself does not have a desirable solubility in vivo and the liposome-based delivery method significantly improved the efficiency of drug delivery. However, the manuscript still presents multiple weaknesses that need to be addressed before the consideration of publication:

1.     Does the encapsulated silymarin directly exert effects in liver or nervous system? Can the drug cross the blood/brain barrier to modulate brain activity?

2.     The description of statistical analysis is largely lacking. For example, Fig. 6a, did you perform a one-way ANOVA with post-hoc tests? This question applies to every bar graph in this paper.

3.     A much more detailed figure legend is needed as it is difficult to interpret the figures without proper figure legends.

4.     Quantification is needed to categorize the cellular morphology of hepatic cells.

5.     Quantification is needed to characterize the pathological changes in the brain

6.     Data of karyolysis is not presented in the manuscript. Please show the associated data.

7.     An additional paragraph about the drug delivery issue of silymarin is needed to set the premise, which should be included in the Introduction.

Author Response

Please see attachemnt

Round 2

Reviewer 1 Report

The authors responded to most of the concerns and corrected the manuscript as requested. It is necessary to check the grammar of the entire manuscript and define BLNPs.

Author Response

Dear Reviewer,

Many thanks for your time, consideration and valuable comments. We are delighted that you are now more satisfied wth our revised work. We also went through the entire manuscript to correct any typos and grammatical mistakes. We also defined the acronym BLNPs.

With many thanks and best regards,

Dr. Menaa (on behalf of all my coauthors)

Reviewer 2 Report

The manuscript has improved significantly after revision. However, the description of statistical analysis still lacks details: if there is no post-hoc, how do you compare different groups with only ANOVA. ANOVA can only tell you the overall difference with no further details. In addition, I still DO NOT see any changes to figure legends: only figure title without anymore descriptions. Please make sure to include the sample numbers and the statistical analysis (like threshold), treatment conditions (concentrations, hours, etc)

Author Response

Dear esteemed Reviewer,

Many thanks again for your valuable comments and suggestions. 

For statistical analysis, 5 samples from each group were taken and compared with another group using independent t test to find out the significance between the two groups being compared. There was a typing error in the previous version of the manuscript for which we apologize. These comparisons were carried out for pairs of different groups such as Normal: Diseased, Diseased: SLNPs, SLNPs: Silymarin, and Zn: ZSLNPs. The p values were recorded as such and values lower than the alpha value of 0.05 were chosen as statistically significant. This information has been included in section “2.5.6. Statistical analysis”, lines 361 to 367.

The figure legends have been altered to include further details about the methodology, sample numbers, statistical analysis, treatment conditions and durations for figures concerning animal studies. For the figures related to characterization of the nanoparticles, more description has been added about the machinery used.   

We hope that your expectations have been fulfilled and we really value your time and appreciate your consideration and feedback which helped us much to improve the manuscript.

With best regards and wishes for 2023!

Drs.  Nosheen F. Rana and Farid Menaa  (on behalf of all our coauthors)